

# A field test of forest canopy structure measurements with the CanopyCapture smartphone application

Christopher H. Lusk

Environmental Research Institute, University of Waikato, Hamilton, New Zealand

## ABSTRACT

**Background:** Several smartphone applications have been developed for the purpose of low-cost and convenient assessments of vegetation canopy structure and understorey illumination. Like standard hemispherical photography, most of these applications require user decisions about image processing, posing challenges for repeatability of measurements. Here I report a test of CanopyCapture, an application that instantaneously estimates percentage canopy gap fraction without any input from the user, and has the added advantage of an intuitive levelling mechanism.

**Methods:** Gap fraction estimates by CanopyCapture were compared with gap fraction values computed by the LAI-2200C Canopy Analyzer, in two contrasting evergreen temperate forests in New Zealand: an even-aged southern beech (*Nothofagus*) stand and an old-growth podocarp/broadleaf forest. These comparisons were repeated using a wide-angle adapter to enhance the smartphone camera's field of view from 45 to 65°. I also asked if CanopyCapture results depended on sky condition (sunny *vs.* overcast) and on the type of smartphone used.

**Results:** CanopyCapture output was significantly correlated with gap fraction computed by the LAI-2200C ($R^2 = 0.39$), and use of the wide-angle adapter lifted this value to 0.56. However, CanopyCapture output was not significantly correlated with LAI-2200C output in the even-aged *Nothofagus* stand, where there was less spatial variation in canopy structure. Despite being much less sensitive to variation in gap fraction than the LAI-2200C, CanopyCapture was nevertheless able to detect differences in average gap fraction between the two forests studied. CanopyCapture results beneath intact canopies were not significantly affected by sky condition, but reflection of direct light off tree trunks in sunny weather caused slight overestimation of gap fraction beneath broken canopies and gaps. Uneven or patchy cloud cover can also cause erroneous readings beneath large canopy openings. Three different models of smartphone gave different results.

**Conclusions:** CanopyCapture offers a rapid and repeatable proxy for comparisons of average canopy gap fraction in multiple stands/forests, provided large sample sizes are used. Measurement under even overcast skies is recommended, and studies involving multiple operators will need to standardize smartphones to ensure comparability of results. Although wide-angle adapters can improve performance, CanopyCapture's low sensitivity prevents high-resolution comparisons of the light environments of individual understorey plants within a stand.

Corresponding author
Christopher H. Lusk,
clusk@waikato.ac.nz

## INTRODUCTION

Ecologists and foresters often have need of high-quality data on forest canopy structure and/or understorey light availability (*Jennings, Brown & Sheil, 1999*). Uses of such data include modelling of seedling survival and growth (*Kobe et al., 1995*; *Montgomery & Chazdon, 2002*), quantifying the distributions of juvenile trees along light gradients (*Lusk, Chazdon & Hofmann, 2006*; *Raymond et al., 2006*), examining evolutionary and plastic responses of plants to shade (*Lusk & Reich, 2000*; *Valladares et al., 2000*; *Kitajima & Poorter, 2010*), managing forestry plantations (*Kim, 1998*), optimizing restoration plantings (*Forbes, Norton & Carswell, 2016*; *Laughlin & Clarkson, 2018*), and investigating forest hydrology (*Díaz, Bigelow & Armesto, 2007*).

The LAI-2200C Canopy Analyzer is a powerful state-of-the-art tool for quantification of canopy structure and understorey light availability (*Danner et al., 2015*; *Pearse, Watt & Morgenroth, 2016*). However, its high cost greatly limits the number of researchers who can benefit from its advantages, especially as working under forest canopies requires the acquisition of two sensors ("wands"), one of which is programmed to take regular measurements in an extensive clearing close to the work site. This requirement for access to a large clearing nearby can also limit the utility of the LAI-2200C in extensive tracts of forest on rugged terrain.

Hemispherical photography obviates the need for access to a clearing and is a more affordable method, especially since the development of digital cameras. The potential for distinguishing between the direct and diffuse components of irradiance offers the additional advantage of very detailed information about understorey light environments (*Chazdon & Field, 1987*; *Rich, 1990*). Hemispherical photography also has several important limitations. Overcast conditions are required for best results, resolution is limited, and determining the optimal exposure and threshold for distinguishing between sky and vegetation can be difficult and prone to user bias (*Rich, 1990*; *Díaz & Lencinas, 2018*; *Chianucci, 2020*). The need to carefully level the camera (and orient it when modelling the solar track), and to process images, can also make hemispherical photography a slow and labour-intensive process. *Arietta (2021)* has shown that smartphone spherical panoramas can ameliorate some of the problems of traditional hemispherical photography.

Densiometers are an affordable low-tech option that can detect broad patterns of spatial and temporal variation in canopy openness (*Strickler, 1959*; *Ganey & Block, 1994*). However, *Cook et al. (1995)* reported that densiometers systematically overestimated canopy cover. Additionally, their low resolution and susceptibility to operator bias renders them unsuitable for projects requiring high precision and repeatability (*Baudry et al., 2014*).

Several free smartphone applications have been developed recently with a view to providing affordable, rapid, and convenient assessments of canopy structure and understorey light availability. These include PocketLAI (*Confalonieri et al., 2013*), Canopeo (*Patrignani & Ochsner, 2015*), GLAMA (*Tichý, 2016*) and Canopy Cover (*Easlon, 2016*). Most require user decisions about image processing, most notably in setting the threshold between "sky" and "vegetation". This significant element of user input, a feature shared with hemispherical photography, poses challenges for repeatability of measurements (*Chianucci, 2020*). CanopyCapture (*Patel, 2018*) is an exception, yielding instantaneous estimates of percentage canopy closure without any user input. Another convenient feature is an intuitive mechanism that helps the operator level the phone before taking a photo. I addressed four questions: (1) Is CanopyCapture output correlated with that of the LAI-2200C Canopy Analyzer? (2) How sensitive is this application to variation in canopy structure? (3) Do results depend on sky condition? (4) Do results depend on the make and model of smartphone utilized?

## MATERIALS AND METHODS

### CanopyCapture

CanopyCapture uses tonal contrast to interpret pixels as "vegetation" or "sky". No user manual or technical specifications have been made available by the developer (*Patel, 2018*), so the thresholding algorithm cannot be reported here. CanopyCapture generates a false colour photo of canopy structure, rendering pixels interpreted as sky in red (Fig. 1). Although percentage "canopy cover" is reported on the screen, what CanopyCapture actually appears to estimate is the complement of percentage gap fraction (100 – gap fraction). *Jennings, Brown & Sheil (1999)* point out that canopy cover (the percentage of the forest floor covered by the vertical projection of tree crowns and other vegetation) is often confused in the literature with canopy *closure i.e.* the percentage of the sky obscured by vegetation when viewed from a singular point, like a camera lens. Moreover, standard smartphone cameras produce rectilinear images, which like any other projection that is not perfectly equisolid, will distort the relative areas of gaps in the canopy as a function of zenith angle (*Fleck, 1995*). Gap fractions must therefore be corrected for area distortion in order to calculate canopy openness, or its complement *i.e.* canopy closure (*Frazer, Trofymow & Lertzman, 1997*). As CanopyCapture does not require any input about lens projections or field-of-view, the figure it reports must be the complement of uncorrected gap fraction (100 – % gap fraction). CanopyCapture output is therefore reported below as "gap fraction". Photos taken with CanopyCapture can only be saved as screen shots, so the "canopy cover" figure must be noted down before taking the next measurement.

### Comparison with LAI-2200C canopy analyzer

Gap fraction estimates by CanopyCapture were compared with gap fraction values computed by the LAI-2200C Canopy Analyzer, in two evergreen temperate forests in New Zealand.

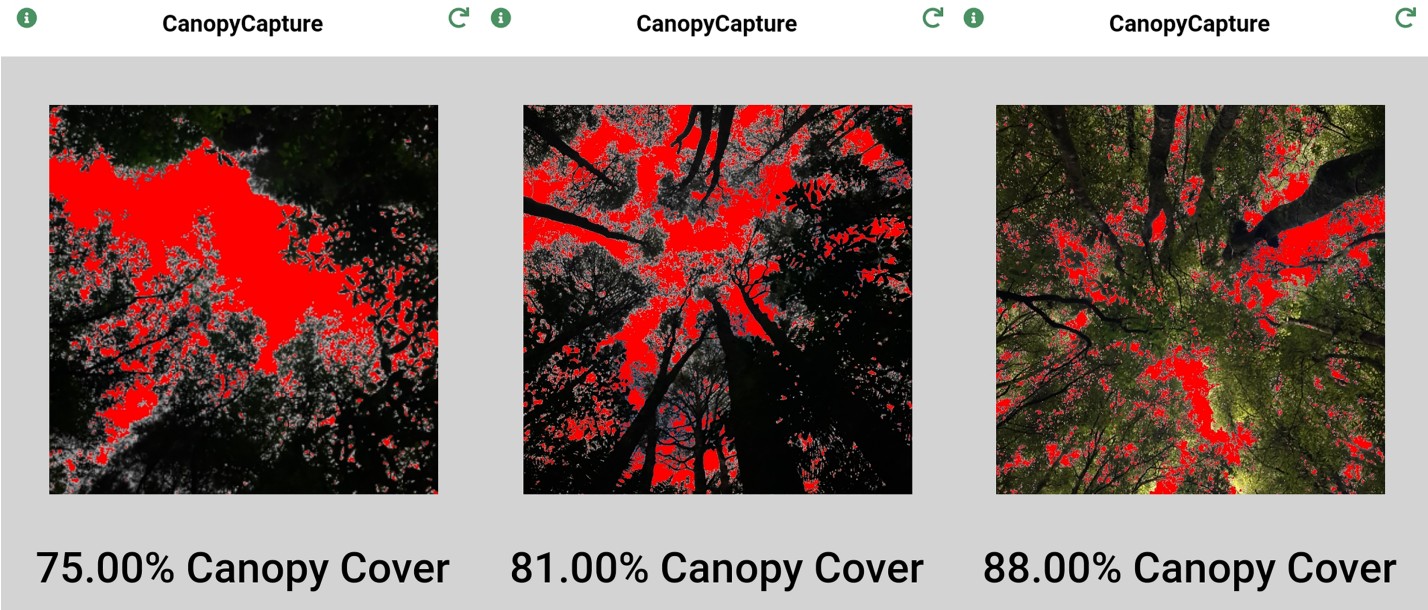

**Figure 1 Canopy structure variation in New Zealand podocarp/broadleaved forest, as represented by the smartphone application CanopyCapture.** A small treefall gap (left), a stand of light-crowned *Dacrycarpus dacrydioides* (centre), and a stand of heavy-crowned *Beilschmiedia tawa* (right).

An old-growth podocarp/broadleaved forest was sampled in the Lake Okataina Scenic Reserve (38.089°S, 176.424°E). The podocarps *Dacrydium cupressinum* and *Dacrycarpus dacrydioides* up to 45 m tall emerge from a canopy dominated by *Beilschmiedia tawa* (Lauraceae) with fewer *Knightia excelsa* (Proteaceae), *Laurelia novae-zelandiae* (Atherospermataceae) and *Litsea calicaris* (Lauraceae). The treeferns *Dicksonia squarrosa* (Dicksoniaceae) and *Cyathea smithii* (Cyatheaceae) are common in the understorey, and sometimes form dense thickets in treefall gaps (*Lusk & Laughlin, 2017*). The monocotyledonous liane *Ripogonum scandens* (Ripogonaceae) forms dense tangles in places. Although some stands were selectively logged during the mid-20th century, most forest in the reserve approximates an old-growth structure, including treefall gaps in varied stages of regeneration. Nomenclature follows *Allan (2002–2021)*. Research at Lake Okataina Scenic Reserve was authorized by the Department of Conservation (66760-RES).

A southern beech (*Nothofagus*) stand was sampled on a poorly-drained site near Mamaku on the Patetere Plateau (38.018°S, 176.067°E). *Nothofagus truncata* up to 22 m tall formed most of the canopy, accompanied by fewer and slightly smaller *N. menziesii*. An approximately normal distribution of *N. truncata* diameters between 15 and 70 cm, and the presence of occasional larger cut stumps in an advanced state of decay, were consistent with establishment of an even-aged *Nothofagus* stand after logging of the previous forest. *Weinmannia racemosa* (Cunoniaceae) and *Ixerba brexioides* (Strasburgeriaceae) were common in the subcanopy and understorey. There were only a few small canopy gaps, resulting from standing deaths of *N. truncata* trees. Access to this forest was provided by Hancock Forest Management (OTPP_91459) and Department of Conservation (97727-RES).

The LAI-2200C quantifies the attenuation of diffuse sky radiation in five angular bands within a quasi-hemispherical field of view, using five sensors arranged in concentric rings. One of two wands was mounted on a tripod and used to take measurements 1.5 m above each sampling point. The other wand was placed in a large clearing that offered an unobstructed view of the sky over the 148° field of view of the instrument, and programmed to take readings at 30 s intervals. Combining data from the two wands enables calculation of diffuse non-intercepted irradiance beneath a forest canopy as a proportion of simultaneous irradiance outside the forest. Computation of gap fraction ($G$) is based on averaging the logarithms of diffuse light transmittance in the five angular bands, weighted by their relative areas:

$$G = e^{\left[ \sum_{i=1}^{5} ln \frac{B_i}{A_i} \ W_i \right]}$$

where the subscript $i$ refers to the five concentric optical sensor rings ($i = 1...5$), $B_i$ is diffuse irradiance reaching the understorey wand from each of the five zenith angle arcs, $A_i$ is diffuse irradiance reaching the clearing wand from each of the same arcs, and $W_i$ is the weighting of each of the five zenith angular bands according to their relative areas.

CanopyCapture installed on a Samsung Galaxy A21s smartphone was used to estimate gap fraction at the same sampling points. The 13 MP front camera used by CanopyCapture has a 45° field of view, meaning it samples a much smaller area of the forest canopy than the LAI-2200C. After each measurement with the LAI-2200C, the phone was placed on the same tripod and CanopyCapture measurements taken with and without a clip-on 37 mm wide-angle HD lens (SourceTon, Guangdong, China) fitted to the phone camera. This lens increased the camera's field of view to 65°. Measurements were made under overcast conditions, between 11:00 and 15:00 h on summer days (3rd January at Okataina, 9th February at Mamaku).

Measurements were taken at 30 points in each forest, randomly spaced 11 to 15 m apart on transects run through the forest. This spacing meant that successive sampling points were unlikely to fall beneath the crown of the same canopy tree, as most crown diameters in New Zealand forests are ≤10 m (*Veblen & Stewart, 1982*; *Walcroft et al., 2005*). All transects were located on flat ground or on slopes of <5°.

## Do results depend on sky condition?

CanopyCapture installed on a Samsung Galaxy A21s smartphone was used to compare canopy closure values obtained under blue and overcast skies. This aspect of the work was carried out in Jubilee Bush (37.775°S, 175.292°E), a 5-ha urban remnant of evergreen temperate forest in Hamilton, New Zealand. The present forest canopy in Jubilee Bush is dominated by *Dacrycarpus dacrydioides* that mostly established after 19th-century logging; and *Beilschmiedia tawa*, some of which are undoubtedly older (*Whaley, Clarkson & Smale, 1997*). There are also scattered larger *D. dacrydioides* (some exceeding 1 m diameter) that predate logging. The other common trees are *Laurelia novae-zelandiae*, *Alectryon excelsus* (Sapindaceae), and *Melicytus ramiflorus* (Violaceae). Recent tree deaths
have left a few small canopy gaps. Research in Jubilee Bush was authorized by Hamilton City Council.

Sampling points were randomly spaced between 11 and 15 m apart along two transects run parallel to the longest axis of the reserve. At each of 27 sampling points, three flat-topped 25 cm plastic stakes were driven into the soil, to provide a level sampling platform 15–20 cm above the forest floor. CanopyCapture measurements were taken from these sampling platforms under overcast and blue skies between 09:30 and 15:00 h in August (winter) 2021.

### Do results depend on make and model of phone?

CanopyCapture installed on three different smartphones was used to estimate canopy closure in a wooded area of the University of Waikato Hamilton campus (37.785°S, 175.316°E). These consisted of a Samsung Galaxy A21s (13 MP front camera with 45° field of view), a Samsung Galaxy S20 (10 MP front camera, 45° field of view), and a Sony H4113 (8 MP front camera, field of view 90°). Fifteen sampling points were spaced at random intervals along a 100-m long transect run along the longest axis of the wooded area. Measurements were made under an overcast sky. The eight different tree species present in the canopy included two deciduous species that were leafless at the time of sampling (August 2021), resulting in wide spatial variation in canopy gap fraction.

### Statistical analysis

The relationships of CanopyCapture output with results from the LAI-2200C were analysed by standardized major axis (SMA), using the *smatr* R package (*Warton et al., 2012*). SMA (rather than ordinary least squares regression) was appropriate here as the scaling of bivariate relationships tells us how sensitive CanopyCapture is to variation in leaf area index and understorey light availability. Variables were log-transformed before some analyses, to improve model fits and normalize distributions of residuals. SMA was also used to compare CanopyCapture measurements carried out under blue and overcast skies.

CanopyCapture measurements made on different smartphones could not be compared by parametric ANOVA, as the distribution of the residuals could not be normalised by any of the attempted transformations. Consequently the non-parametric alternative of Friedman's test (*Friedman, 1937*) was used to compare means, by way of the *agricolae* R package (*Mendiburu, 2015*). Variances of CanopyCapture values obtained using the three phones were compared with Levene's test for homogeneity of variance, using the *car* R package (*Fox et al., 2012*).

## RESULTS

### Comparison with LAI-2200C canopy analyzer

CanopyCapture output was significantly correlated with gap fraction computed by the canopy analyzer in the old-growth podocarp/broadleaf forest at Okataina ($R^2 = 0.39$), and use of the wide-angle adapter lifted this figure to 0.56 (Fig. 2A). Use of the wide-angle adapter affected neither the slope nor the elevation of the standardized major axis (SMA)

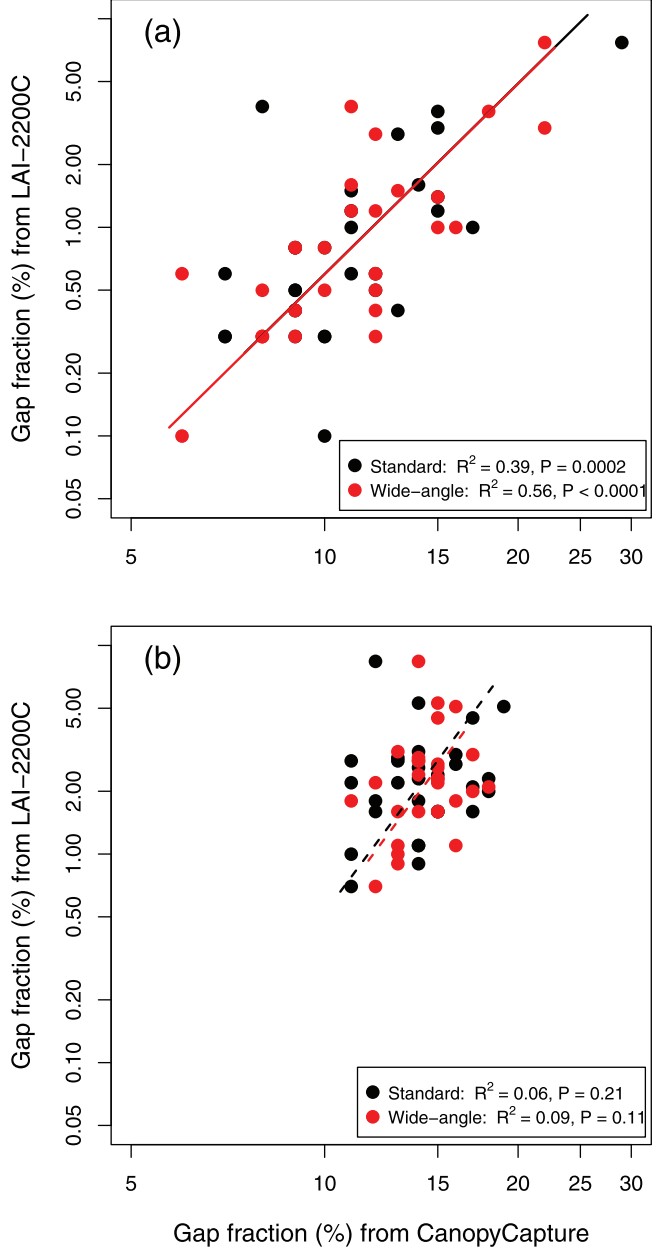

**Figure 2 Standardized major axis relationships of CanopyCapture output (with and without wide-angle adapter) with gap fraction computed by the LAI-2200C, in two New Zealand forests.** (A) Old-growth podocarp/broadleaved forest at Okataina. Standard smartphone camera lens: SMA equation $y = 0.00044x^{3.13}$. Wide-angle adapter: SMA equation $y = 0.00064x^{2.97}$. (B) Even-aged *Nothofagus* stand near Mamaku. Standard smartphone camera lens: SMA equation $y = 0.00022x^{3.47}$. Wide-angle adapter: SMA equation $y = 0.0000044x^{4.92}$.

of these relationships (Fig. 2A). Slopes of ~3.0 show that CanopyCapture has low sensitivity to spatial variation in canopy gap fraction (Fig. 2A): the SMA equations indicate that a doubling of gap fraction as measured by CanopyCapture equated to an eight- to nine-fold increase in gap fraction as measured by the LAI-2200C at Okataina.

**Table 1 Means comparisons of canopy gap fraction (%) in old-growth podocarp/broadleaf forest at Okataina and even-aged *Nothofagus* forest at Mamaku, as estimated by the LAI-2200C Canopy Analyzer and the CanopyCapture smartphone application.** All variables were log-transformed before analysis to improve normality, and so the statistics shown below are geometric means and the back-transformed span of one standard deviation.

| Site | CanopyCapture | | LAI-2200C |
| --- | --- | --- | --- |
| | Standard | Wide-angle | |
| Okataina | 11.0 (8.1–14.9) | 11.0 (8.0–15.1) | 0.8 (0.3–2.1) |
| Mamaku | 14.1 (12.0–16.5) | 14.4 (12.8–16.1) | 2.2 (1.2–3.8) |
| *T*-test *P* value | 0.0002 | 0.0001 | <0.0001 |

CanopyCapture output was not significantly correlated with LAI-2200C output in the even-aged *Nothofagus* stand at Mamaku (Fig. 2B), where there was less spatial variation in canopy structure (Table 1). Although use of the wide-angle adapter raised the $R^2$ value by >50%, the relationship remained non-significant (Fig. 2B).

Despite the relatively low sensitivity of CanopyCapture to spatial variation in canopy structure, it did nevertheless detect significant differences in average gap fraction between the two forests—both with and without the wide-angle adapter (Table 1). CanopyCapture did, however, find proportionally smaller means differences than the LAI-2200C (Table 1).

### Do results depend on sky condition?

CanopyCapture measurements under blue skies scaled as approximately $1.195x-17.0$, where $x$ = gap fraction estimated under an overcast sky (Fig. 3); 95% confidence intervals of the slope did not include unity (1.003–1.423), providing conclusive evidence of a departure from isometry. Under dense canopies, sky condition made little or no difference to gap fraction estimates; under more broken canopies, gap fraction values recorded under blue skies tended to be slightly higher than those recorded in overcast conditions.

### Do results depend on make and model of phone?

Friedman's test indicated that mean gap fraction estimates differed significantly between phones ($F = 10.55$, $P = 0.0004$), and post-hoc pairwise comparisons showed the mean value recorded on the Samsung A21s was significantly higher than those obtained with the other two phones (Fig. 4). Levene's test indicated homogeneity of variances across the three phones ($F (2, 42) = 0.17$, $P = 0.85$), despite the narrower range of values obtained with the Sony H4113 phone (Fig. 5).

## DISCUSSION

CanopyCapture was able to distinguish between two forests that differed significantly in average gap fraction (Table 1). Leaf area index and understorey light levels vary widely in relation to climate, soils and successional development (*Luo et al., 2004*; *Brantley & Young, 2007*; *Iio et al., 2014*), as well as being influenced by more local differences in crown depth and density of canopy tree species (*Canham et al., 1994*; *Fritz & Lusk, 2020*). CanopyCapture appears suitable for detecting such differences among forests or stands
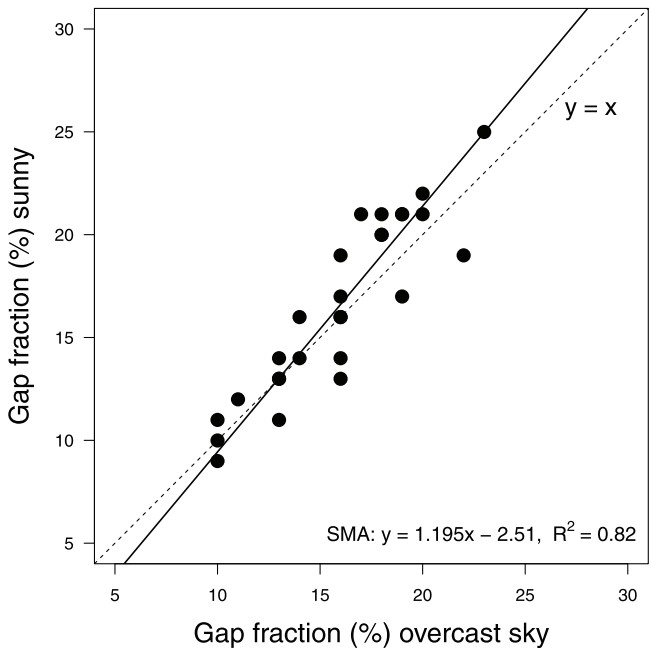

**Figure 3 Scaling of canopy closure measurements by CanopyCapture at the same 27 random points under overcast and clear skies, in a temperate evergreen forest in New Zealand.** Regression line shows standardized major axis of relationship. Dashed line shows $y = x$.

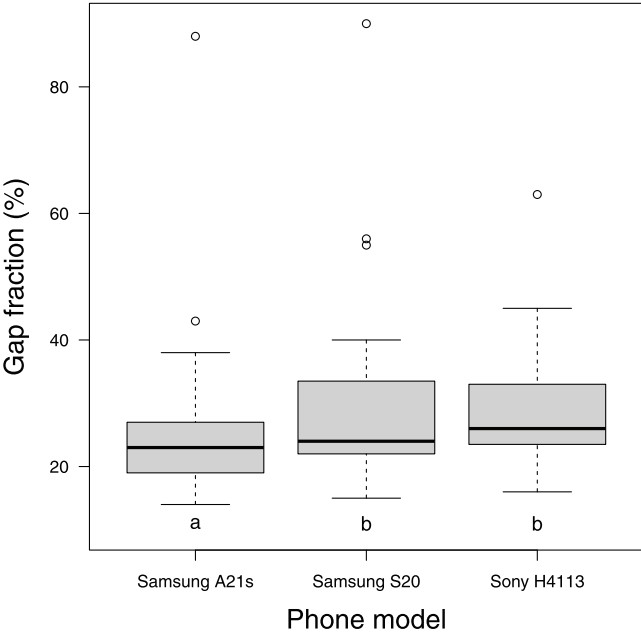

**Figure 4 Distribution of gap fraction measurements with CanopyCapture on three different models of smartphone, at the same 15 random points beneath a wooded area of University of Waikato Hamilton campus.** Mean gap fraction closure values do not differ significantly between phones that share the same letter.

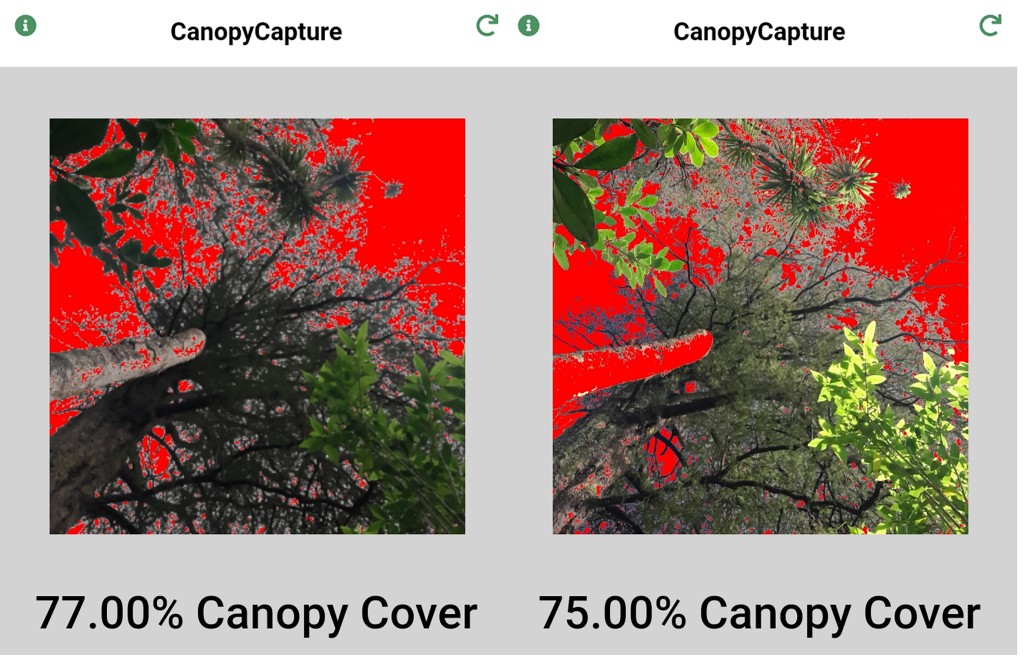

**Figure 5 CanopyCapture photographs taken at the same point under overcast (left) and sunny skies (right).** Note that reflection of sunlight causes a tree trunk to be misread as sky under sunny conditions, resulting in slight overestimation of gap fraction.

(*e.g.*, monitoring canopy closure during forest restoration projects), with the caveat that large numbers of points should be sampled to compensate for the narrow field of view and low sensitivity.

CanopyCapture output proved a less reliable proxy for spatial variation in canopy structure within a stand, primarily because of its low sensitivity. CanopyCapture's tendency to darken the more open parts of a forest and to brighten the shadier microsites indicate that the application does not override the camera's automatic settings that adjust photograph exposure by normalizing the luminance histogram of images (cf. *Macfarlane et al., 2014*). CanopyCapture responded well to the typically wide range of local canopy closure and understorey illumination found in the old-growth forest at Okataina, especially when the smartphone camera's field of view was enhanced with a wide-angle adapter (Fig. 2A). However, CanopyCapture lacked sufficient sensitivity to accurate represent the more restricted variation in canopy structure in the even-aged stand at Mamaku, where the canopy analyzer indicated only about one order of magnitude of variation in gap fraction (Fig. 2B) *cf*. nearly two orders in the old-growth forest at Okataina (Fig. 2). This low sensitivity means CanopyCapture cannot provide a high-resolution proxy for the light environments of individual understorey plants within a stand, as is often needed in studies of the influence of light on growth, survival, or functional traits (*e.g. Kobe et al., 1995*; *Lusk & Reich, 2000*; *Montgomery & Chazdon, 2002*).

CanopyCapture is best used under overcast conditions if there are conspicuous gaps in the canopy. Reflection of direct light off smooth, pale-coloured tree trunks was one source of error in sunny conditions (Fig. 6), these reflective surfaces sometimes being incorrectly

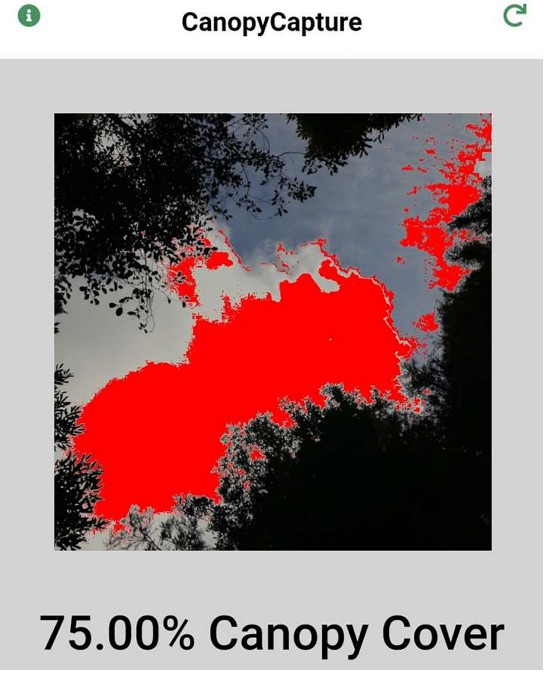

**Figure 6  Patchy cloud cover above an extensive canopy opening causes misread of parts of the sky as canopy, resulting in gross underestimation of gap fraction.**

interpreted as "sky" in places where canopy openings admitted broad sunbeams. Hemispherical photography in sunny conditions is subject to the same problem, requiring manual darkening of reflective surfaces that appear brighter than the sky in the original image (*Li et al., 2020*); although use of smartphone spherical panoramas can minimize this problem (*Arietta, 2021*). Reflection of direct light off tree trunks may be the main reason why in the more open parts of the stand, gap fraction estimates recorded under blue skies tended to be slightly higher than those recorded under overcast conditions (Fig. 3). At gap fractions <15%, overhead conditions made little or no difference to CanopyCapture output, suggesting weather is less important for measurements made under intact canopies with few obvious gaps.

Although not influencing the results presented in this paper, spatial variation in the luminosity of the sky can cause large errors in CanopyCapture measurements beneath large canopy openings. If most of the field of view is occupied by sky, and if cloud cover is patchy or of variable density, the darkest regions of the sky can be misread as canopy, resulting in gross underestimates of gap fraction (Fig. 6). Like the problem of reflection from tree trunks (Fig. 5), this artefact means that CanopyCapture performs most reliably beneath canopies lacking large gaps; it also highlights the importance of working under uniform overhead conditions.

Significant discrepancies between CanopyCapture results obtained with different phones (Fig. 4) show that a standard phone model must be used in multi-operator studies. Differences in field of view (45° *vs.* 90°) are one likely cause of discrepancies between

different phones; however, this factor is not a sufficient explanation, as mean estimated canopy closure differed most clearly between two Samsung phones with cameras that shared the same field of view but differed in resolution (Fig. 5). Differences in camera software routines may also influence results obtained from photographic applications. Canopy structure assessments by another smartphone application (Canopeo: *Patrignani & Ochsner, 2015*) have also been found to depend on the model of phone used (*Heinonen & Mattila, 2021*).

## CONCLUSIONS

CanopyCapture offers a rapid and repeatable proxy for comparisons of average canopy cover and/or understorey illumination in multiple forest stands, provided large sample sizes are obtained from each stand. The repeatability resulting from the lack of user input suits CanopyCapture well to studies integrating data obtained by multiple users working in different locations, provided all participants use the same model of smartphone.

The intuitive levelling mechanism is also convenient, obviating the need for the operator to carry a tripod. Although enhancement of smartphone cameras' field of view with wide-angle or fisheye adapters has the potential to improve performance, CanopyCaptures's low sensitivity prevents high-resolution comparisons of the local light environments of individual plants within a stand.

## ACKNOWLEDGEMENTS

I thank Danielle Le Lievre and Bibishan Rai for field assistance, Kiri Wallace for suggesting use of smartphone applications for assessing canopy structure, and the reviewers for their constructive comments.

### Funding

This work was funded by the Royal Society of New Zealand through Marsden grant 20-UOW-041. The funders had no role in study design, data collection and analysis, decision to publish, or preparation of the manuscript.

### Grant Disclosures

The following grant information was disclosed by the authors:
Royal Society of New Zealand: 20-UOW-041.

### Competing Interests

The author declares he has no competing interests.

### Author Contributions

Christopher H. Lusk conceived and designed the experiments, performed the experiments, analyzed the data, prepared figures and/or tables, authored or reviewed drafts of the paper, and approved the final draft.

## Field Study Permissions

The following information was supplied relating to field study approvals (*i.e.*, approving body and any reference numbers):

Work at both sites was authorized by Department of Conservation. Access to the forest near Mamaku was provided by Hancock Forest Management Ltd.

## Data Availability

The measurements taken using the smartphone application and the LAI-2200C canopy analyzer are available in the Supplemental File.

## Supplemental Information

Supplemental information for this article can be found online at http://dx.doi.org/10.7717/peerj.13450#supplemental-information.

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
