# Peer review of "A field test of forest canopy structure measurements with the CanopyCapture smartphone application"

_PeerJ, doi:10.7717/peerj.13450_

## Round 0.1 · original submission · Major Revisions

Would you please address the comments of reviewers 1 and 2 point-by-point?

Reviewer 1 ·

Basic reporting

No comment

Experimental design

1. Comparing this App with other Apps
This is the most efficient way to show how good this App is.

2. Canopy closure, canopy cover and leaf area index (LAI) are three different canopy properties.
Canopy closure (=1- canopy openness), canopy cover and leaf area index are related to each other, but they are not identical (Chen et al. 1997; Jennings et al. 1999). Canopy openness shouldn’t be used interchangeably with canopy cover (in Figures 2 and 4). It is improper to use LAI (acquired with LAI-2200s as a benchmark) to evaluate canopy cover of CanopyCapture. Since the confusion between canopy closure and canopy cover is ubiquitous, and canopy closure (rather than canopy cover) is the most direct result of canopy image analysis, it is better to confirm with the CanopyCapture developer which canopy property is actually estimated by this APP. In the indirect methods (such as hemispherical photography and LAI-2200c) for measuring light availability, percent light availability is derived from gap fraction (LI-COR Biosciences 2021; Rich 1990). Before revising this manuscript, it is strongly suggested to understand the definitions of canopy closure, canopy cover, leaf area index, gap fraction and light availability.

3. The field of view should be controlled.
The poor performance in Figure 2b and Figure 4 will discourage the use of this App. The poor relationships may be attributed to that the field of view of LAI-2200 is much higher than that of smartphone cameras. Using a wide-angle adapter (such as https://www.amazon.com/Camera-Adapter-SourceTon-Samsung-Smartphones/dp/B07415BD9B) can increase the field of view of smartphones. Hopefully, this approach can raise the performance.

4. Using RMSE rather than R-squared
Please use RMSE (root mean square error) rather than R-squared to evaluate the performance of this App. R-squared is to quantify the extent that a dependent variable is explained by a regression model. RMSE is to quantify the differences between two variables (in this manuscript, variable measured with CanopyCapture and variable measured with LAI-2200s) (Confalonieri et al. 2013).

5. The problem of being unable to measure light availability > 5% should be solved.
The understorey light availability of many vegetation types (such as young plantations, forests recovering from clearing ) is higher than 5% (Beaudet et al. 2004). The existing methods (e.g. hemispherical photography, LAI-2200s, quantum sensors, GLAMA app) don’t have such problem. The use of this App will be greatly limited if this problem isn’t solved.

6. The misclassification problem of canopy and sky should be solved.
This problem was shown in Figures 6 and 7 and will considerably reduce the performance of this App in a high light environment. This problem can be mostly eliminated through thresholding, but this App doesn’t include this function. Please elucidate whether this problem can be solved through any approach.

References
I. Beaudet, M., Messier, C., and Leduc, A. (2004). "Understorey light profiles in temperate deciduous forests: recovery process following selection cutting." J. Ecol., 92(2), 328-338.
II. Chen, J. M., Rich, P. M., Gower, S. T., Norman, J. M., and Plummer, S. (1997). "Leaf area index of boreal forests: Theory, techniques, and measurements." Journal of Geophysical Research-Atmospheres, 102(D24), 29429-29443.
III. Confalonieri, R., Foi, M., Casa, R., Aquaro, S., Tona, E., Peterle, M., Boldini, A., De Carli, G., Ferrari, A., Finotto, G., Guarneri, T., Manzoni, V., Movedi, E., Nisoli, A., Paleari, L., Radici, I., Suardi, M., Veronesi, D., Bregaglio, S., Cappelli, G., Chiodini, M. E., Dorninoni, P., Francone, C., Frasso, N., Stella, T., and Acutis, M. (2013). "Development of an app for estimating leaf area index using a smartphone. Trueness and precision determination and comparison with other indirect methods." Comput. Electron. Agric., 96, 67-74.
IV. Jennings, S. B., Brown, N. D., and Sheil, D. (1999). "Assessing forest canopies and understorey illumination: canopy closure, canopy cover and other measures." Forestry, 72(1), 59-73.
V. LI-COR Biosciences (2021). LAI-2200C- Plant Canopy Analyzer instruction manual, LI-COR Biosciences, Lincoln, Nebraska, USA.
VI. Rich, P. M. (1990). "Characterizing plant canopies with hemispherical photographs." Remote Sens. Rev., 5, 13–29.

Validity of the findings

No comment

Additional comments

1. There is no Figure 3. The figure below Figure 3 is Panel b of Figure 2.
2. Putting the line of X=Y in Figure 4 will show whether the relationship of the two variables varies along the gradient of canopy cover.
3. In Figure 2, the dots were round, and the legends were square. Please be consistent.
4. In Table 1, standard deviations should be provided.
5. In the caption of Figure 2, please mention that the X and Y axes are both in log scale.
6. L 175-176: The result (Figure 2a) is not enough to support this conclusion.

Reviewer 2 ·

Basic reporting

This is a useful comparison and validation of a smartphone canopy measurement application (CanopyCapture) that might be an alternative to more traditional methods. The author makes three comparisons: 1.) They compare the canopy closure estimate from CanopyCapture app with ‘light availability’ from a LiCore LAI-2200c instrument. They also compare canopy cover from CanopyCapture 2.) across clear or overcast days and 3.) among phone models.
The statistical methods are sound and well-reasoned. Overall, the writing is clear.
I have two primary concerns with the study. First, I think the framing is spurious. Second, I have a hard time evaluating the results of the comparison between the LAI-2200 and CanopyCapture because it seems two different measurements are used.

Experimental design

One concern is in the comparison between LAI-2200c “light availability” and CanopyCapture’s “canopy cover.” It is not clear from the methods section what exactly is being estimated with “light availability.” It would be useful to explain this clearly and perhaps provide the equation from which it is derived in order to know if a comparison with canopy cover is justified. Because hemispherical projections like LAI-2200 measure canopy CLOSURE, not canopy COVER, it is confusing (see Jennings, S. B., Brown, N. D., and Sheil, D. (1999). Assessing forest canopies and understory illumination: canopy closure, canopy cover and other measures. Forestry 72, 59–74.). All estimates of LAI, canopy cover, closure, and (I assume) light availability are based on estimates of gap fraction, so it seems that gap fraction would be a better statistic for comparison.

Validity of the findings

The framing of the introduction sets up the relative advantage and disadvantages of canopy measurement methods as a comparison of the LAI-2200, the CanopyCapture app, hemispherical photography (HP), and densiometers. However, these aren’t mutually exclusive methods and some of the contrasts are spurious. The LAI-2200c is essentially just a small hemispherical camera in which all estimates are computed under-the-hood. Treating them as separate methods is misleading. For instance, HP also requires calibration from a clearing or second sensor (line 52-53) and the LAI-2200c also requires overcast conditions (line 56-57, lines 229-231) unless direct sun is blocked and additional measures of light scatter are acquired (https://www.licor.com/env/support/LAI-2200C/topics/sun.html) (which is also true for HP). Similarly, both HP and LAI-2200c need to be leveled (line 58). The author mentions that HP needs to be carefully oriented (line 58) but this is only true if one is plotting sun paths. For estimates relying solely on gap fraction (like canopy closure and LAI) orientation doesn’t matter for HP, either. I believe that the author needs to more clearly articulate exactly where the advantages and disadvantage lie for different methods while being careful not to conflate methodological difference origination from using the tools to measure different things (i.e. light environment like Site Factors versus canopy structure like gap fraction or canopy cover).
Along the same lines, I think that the author needs to address the fact that not all HP methods are the same. The disadvantages of traditional methods of HP with SLR cameras (as in Rich 1990) differ from those using single-shot estimates from smartphones (as in Bianchi et al 2017) and many problems have been obviated by HP generated from spherical panoramas captured on smartphones (as in Arietta 2021).
The framing that thresholding “normally involves some degree of subjectivity” (line 69) with Canopy Capture as an objective exception (line 71) is misleading. All of the comparisons that the author cites (line 68) use objective, algorithmic thresholding. Tichy (2016) and Arietta (2021) use histogram-based threshold algorithms. Patrignani & Ochsner (2015) use automatic color-based thresholds. Bianchi et al (2017) use both. The only difference is that all of this happens in a “black box” with CanopyCapture, but that isn’t a question of more or less objective.

Additional comments

Line 66-69. I think it would be useful to summarize the smartphone applications available. Of the citations listed for this sentence, only Buchi et al (2018) present a smartphone app (GLAMA); the others are all methods that use smartphones but are not smartphone apps themselves.

Line 72-73. I don’t think that the leveling mechanism is very important. Hand-leveling doesn’t introduce noticeable bias in traditional HP (see: Origo, N., Calders, K., Nightingale, J., and Disney, M. (2017). Influence of levelling technique on the retrieval of canopy structural parameters from digital hemispherical photography. Agric. For. Meteorol. 237-238, 143–149.) and Arietta’s (2021) method with smartphone spheres obviates the need to level at all.

Line 127-152. These two parts of the study are sound and I really appreciate that the authors included them. These analyses are important in understanding the limits of this method.

Line 153-166. The statistical analyses are appropriate. I especially think that the use of SMA rather than OLS is wise.

Line 244-251. In addition to hardware differences between phones, it is important to mention that software difference likely play a much larger role. In contrast to traditional cameras, smartphones rely on computational photography to capture images. The difference in software routines has a much larger effect than difference in hardware.

Line 252-260. The conclusions make good sense. I agree with the author’s appraisal.

The figures and table look good. I especially appreciate the example outputs from CanopyCapture.

Reviewer 3 ·

Basic reporting

- Clear and unambiguous professional English used throughout the manuscript.
- Literature references and sufficient field background/context not provided. I have specific comments on this this in the attached document.
- I don't see any raw data shared but the author maintained professional article structure and tables/figures.
- I don' think the application/software of interest has the merit to be the centre point of a scientific article. However, the author asked good research questions and presented data accordingly.

Experimental design

I have a lot of questions and concerns related to the experimental design. I have shared them in the attached document in the relevant sections.

Validity of the findings

I am not convinced that the overall merit of this article is novel. I have made several comments in the relevant sections.

Annotated reviews are not available for download in order to protect the identity of reviewers who chose to remain anonymous.

---

## Round 0.2 · Major Revisions

Thank you very much for revising your manuscript. However, reviewer 1 still has a number of comments you should address.

Please benchmark the CanopyCapture against a reference method, using a suitable and comparable parameter.

And please make sure to include enough information in the text (not only the rebuttal letter) answering the remaining doubts.

Reviewer 2 ·

Basic reporting

I appreciate the author’s effort in revising the manuscript. I especially think that the additional analysis with a wide-angle lens adapter strengthens the study. However, I have major concerns that terminology is being used incorrectly and the relative strengths and weaknesses of different approaches are somewhat opaque and misleading.

Experimental design

I understand that the author’s purpose is to test how well CanopyCapture compares to the industry-standard LAI-2200C instrument. However, this is only possible if the same metrics are compared. As written, it is impossible to determine if this is the case. Despite all three reviewers pointing out the issues comparing very different instruments and very different metrics, the author has not addressed this sufficiently in the manuscript.

Validity of the findings

In their rebuttal to the initial review, the author states, “I wanted to see if CanopyCapture measurements can serve as a proxy for actual measurements of (a) leaf area index and (b) understorey illumination, which is why I report correlations with those variables and not gap fraction.” However, the LAI-2200C are also just a proxy for LAI and light, not “actual measurements” which would be far more difficult. Since both methods are using Gap Fraction estimates to calculate proxy measures of LAI and light, it makes far more sense to compare Gap Fraction. (See comments on specific lines below).

In any case, because misusing terms can lead to a lot of confusion, the author needs to clearly define the metrics that they are using. Again, I think it would be useful to explain this clearly and perhaps provide the equations from which the metrics are derived in order to know if a comparison with canopy cover is justified. This could easily go into the appendix and referenced in the methods section.

Additional comments

Lines 54-55. This statement is factually incorrect. HP also requires calibration from a clearing for accurate measurements. Many practitioners use an estimate (either from a single image from a clearing or from point estimates within the stand) rather than simultaneously capturing two images, but the same practice can be used with LAI-2200C to avoid the second sensor.

Lines 61-63. This statement is factually incorrect. Not all HP based methods require exogenous leveling. Arietta’s (2021) method automatically levels the hemispherical images. Perhaps add this as a caveat.

Lines 88-91. I’m not sure if the app is measuring canopy closure either. When canopy closure is measured with an optical device with a projection other than perfect equasolid angle, it has to be corrected for zenith angle. This is because the gap area is distorted by the lens projection. Since the CanopyCapture app doesn’t seem to require any input about lens projections or field-of-view, the estimate must be uncorrected Gap Fraction. Frazer (1997) explains this well: “gap fractions must be corrected for area distortion before calculations of canopy openness or cover are made. Percent open sky is defined as the percentage area of the sky hemisphere that is unobstructed by vegetation, and can be calculated as the sum of all area corrected gap fractions multiplied by 100% (ter Steege 1993).”

Lines 129-134. I appreciate that the author included the additional analysis with the wide-angle lens adapter. This strengthens the manuscript in my opinion. Was the projection of the adapter/lens combination calibrated and used to correct the area distortion for Canopy Closure estimates? If not this need to be stated. Also, if not, then the metric is uncorrected Gap Fraction, not Canopy Closure.

Lines 228-229. I think this sentence implies that the phone camera adjusts exposure according to the average brightness of the field of view? That is not how any modern cell phone made in the last decade works. Cell phones use computational photography and HDR to adjust individual pixel to normalize the luminance histogram of the image. Only older single-shot point-and-shoot or (sometimes) DSLRs adjust overall exposure based on average brightness.

Lines 243-244. This is largely obviated by HP methods derived from smartphone images (as in Arietta, 2021) because those methods expose for restricted portions of the viewing field at a time and utilize the camera’s built-in HDR methods.

---

## Round 0.3 · accepted · Accept

Thank you very much for your excellent contribution!

Reviewer 2 ·

Basic reporting

I appreciate that the author persisted through these reviews. I think the final product is very good and a useful addition to the literature.

Experimental design

With the canopy metrics of comparison now clarified, the experimental design is warranted and appropriately reported.

Validity of the findings

No comment.

Additional comments

None.